# Early coauthorship with top scientists predicts success in academic careers

Weihua Li [1,2], Tomaso Aste[1,2], Fabio Caccioli[1,2,3] & Giacomo Livan[1,2]*

We examined the long-term impact of coauthorship with established, highly-cited scientists on the careers of junior researchers in four scientific disciplines. Here, using matched pair analysis, we find that junior researchers who coauthor work with top scientists enjoy a persistent competitive advantage throughout the rest of their careers, compared to peers with similar early career profiles but without top coauthors. Such early coauthorship predicts a higher probability of repeatedly coauthoring work with top-cited scientists, and, ultimately, a higher probability of becoming one. Junior researchers affiliated with less prestigious institutions show the most benefits from coauthorship with a top scientist. As a consequence, we argue that such institutions may hold vast amounts of untapped potential, which may be realised by improving access to top scientists.

---

[1] Department of Computer Science, University College London, London WC1E 6EA, UK. [2] Systemic Risk Centre, London School of Economics and Political Sciences, London WC2A 2AE, UK. [3] London Mathematical Laboratory, 8 Margravine Gardens, London WC 8RH, UK. *email: g.livan@ucl.ac.uk

For a number of decades, the growing availability of data about published research has increasingly empowered academia to study itself. Starting from the pioneering work on citation indexes by Garfield in the 1950s[1] and on citation networks by de Solla Price in the 1960s[2], the study of published research developed into the field of bibliometrics[3], which is concerned with the quantitative analysis of all aspects of academic life[4] and with the development of metrics to quantify the quality and impact of researchers[5,6], journals[7], and academic institutions[8]. Recently, thanks in part to the availability of increasingly rich datasets, bibliometrics has witnessed new interest coming from the multidisciplinary data science community, whose research outputs in the field have often been referred to as "science of science"[9–11].

A long-standing theme in bibliometrics is the quantification of the academic impact of individual researchers, which has been the subject of countless studies throughout the years[12–14]. Academic impact is a complex and multifaceted concept. Yet, from an operational point of view nowadays it is increasingly equated to a scientist's ability to attract large numbers of citations. This, in turn, is mainly due to the fact that citations are reliably and consistently recorded across several disciplines, and, most importantly, to the fact that citation-based bibliometric indicators are often used as metrics to rank scholars and determine their career advancements[15]. Despite considerable controversy (see, e.g. [16]), such practices are widely adopted, and therefore the vast majority of studies in the literature use citation-based metrics as a proxy for academic impact[9].

A relevant theme within the devoted literature is that of identifying early indicators of long-lasting academic impact and their manifestation in a junior researcher's career (see, e.g. [9,17]). This is a notoriously challenging task, since the aspects of a researcher's career that are less difficult to quantify do not necessarily yield a large predictive power. Indeed, the productivity of most scientists fluctuates heavily over time[18,19]. Moreover, the unpredictability of the occurrence of a scientist's "greatest hits" over their career trajectory[14] complicates the matter even further.

Due to such challenges, a number of studies in recent years took a different approach by seeking to predict academic impact based on the visibility of a junior researcher[20,21]. Quantifying visibility presents its own challenges, as it encompasses a number of semi-qualitative aspects that contribute to provide a junior researcher's output with a competitive advantage with respect to output of the same quality published by peers with similar academic status and seniority. Factors that contribute to the visibility of a junior researcher are the following: (i) the journals where her research is published[22,23], (ii) the prestige of the institutions she and her coauthors are affiliated with[24], and (iii) the reputation of her more established coauthors[25] and, more generally, her academic social network[26,27].

Multiple approaches have been proposed in the literature to quantify the first two factors above, resulting in the development of indices aimed at measuring the impact and prestige of journals and institutions. In the former case, the most popular option is the impact factor[7], whose use has however attracted considerable criticism over the years[28], which in turn has spurred several alternative proposals[29]. Similarly, university rankings have become a fundamental-yet-controversial element of academic decision making[30,31]. Despite the debate around them, these measures offer practical solutions to assess the long-term impact that such otherwise intangible factors have on an academic career. For example, it has been shown that the academic prestige of the institution(s) a junior researcher is affiliated with correlates positively with long-term impact, as it leads to higher productivity[32] and a higher probability of securing a tenured position

and, more generally, of ending up in a more influential position within a discipline[33,34].

The third factor in the above list, i.e. the "social factor" that contributes to a researcher's visibility, is even harder to quantify. Shortly after its inception, bibliometrics recognized that the development of scientific knowledge hinges to some extent on a "sociology of science"[35], i.e. on the networks of collaborations, interactions, and social relationships that underpin the scientific community. In recent years, this stream of research has enjoyed renewed attention, thanks to the mining of increasingly detailed datasets documenting interactions between scientists on various levels, with studies showing, for example, how academic networks improve the predictability of academic success[26] and have an impact on the speed[36] and likelihood of publication[37] in journals.

Quantifying the aforementioned social factor is especially challenging in the case of junior researchers, whose academic social network is still relatively sparse compared with that of established scientists. A number of studies have circumvented this problem by restricting a junior researcher's social network to her mentors and supervisors, showing in general that the supervision of an impactful mentor has beneficial effects on a protégé's academic career[38,39]. In a similar spirit, a recent paper has revealed a "chaperone effect"[40] in scientific publishing, showing that publishing in high-impact venues as a senior author is exceedingly more likely for scientists who have already done so in the past as junior researchers.

The above body of work suggests that the protracted interaction between a junior researcher and a well-established senior collaborator has long-lasting positive effects. In this paper, we take this body of literature to its extreme consequences, and ask whether single events of interaction with top scientists can have career-altering effects on a junior researcher's future. Our main claim is that the mere coauthorship with a top scientist leads to a lasting competitive advantage in terms of impact. We demonstrate this by means of a matched pair experimental design, splitting a large pool of authors with long-lived academic careers into two groups—those who coauthored at least one paper with a top-cited scientist early on in their career and those who did not. We show that—all other things being equal—junior researchers belonging to the former group enjoy a persistent competitive advantage with respect to their peers belonging to the latter, which ultimately results in a much better chance of becoming top-cited scientists themselves.

In the following, we show the presence of such a competitive advantage for junior researchers across four different scientific disciplines, and we demonstrate the robustness of this finding after controlling for a number of potential confounding factors. Finally, we also show that this result yields significant predictive power, as it can be exploited to improve the predictability of a junior researcher's long-term academic impact based on their early career indicators.

## Results

**Definitions.** Let us begin by introducing the operational definitions of top scientist and junior researcher that we shall retain throughout the rest of the paper. We say that a researcher is a top scientist in a given year if she belongs to the top 5% of cited authors in her discipline for that same year. Such a choice is dictated by the need to find a reasonable balance between the numbers of top and non-top scientists in our following analyses. Furthermore, this choice leads to significant stability in our classification, as in more than 95% of cases in our dataset, once a researcher becomes a top scientist she remains one until the end of her career.

We then classify as junior researchers scientists who are in their first 3 years of academic activity. More precisely, we classify a scientist as a junior researcher for the first 3 years since her first publication, which we reasonably expect to roughly cover the duration of a Ph.D. The main results presented in the following are qualitatively unchanged when extending such period to the first 5 years after the publication of the first year.

**Institutional prestige and impact.** We begin our analysis by pooling together all researchers from four disciplines (Cell Biology, Chemistry, Physics, and Neuroscience—see "Methods" section and Supplementary Tables 1–4 for the lists of journals we consider for each discipline) whose career started between 1980 and 1998 and lasted at least 20 years, who have at least ten publications, and who have published at least one paper every 5 years. In total, we have 22,601 such researchers (see Table 1 for a detailed breakdown in terms of disciplines). Within such pool of authors with long-lived careers, the unconditional probability of being a top scientist in the 20th career year is 24.8%. Let us now proceed to condition this probability based on the institutional prestige a junior researcher is embedded in. We assign an institutional prestige score to each junior researcher in the dataset in order to generate a continuous prestige spectrum, which will allow us to analyze individual career trajectories at a granular level. We do so by means of the average adjusted Nature Index (see "Methods" section) of the researcher's institution and of the institutions her coauthors are affiliated with. We cross-check such a score by computing the Kendall correlation between the ranking of the institutions based on their Nature Index and the widely recognized Leiden ranking[8], getting a correlation coefficient of 0.98 for Cell Biology, 0.94 for Chemistry, 0.94 for Neuroscience, and 0.97 for Physics, respectively.

In Fig. 1 (left panel) we report the number of junior researchers falling within each quintile of the institutional prestige distribution, divided into three groups: those who did not coauthor a paper with a top-cited scientist early in their career (15,495 authors, shown in blue), those who coauthored papers with one top-cited scientist (4573 authors, shown in orange), and those who coauthored papers with more than one top-cited scientist (2533 authors, shown in red). In Supplementary Fig. 1 we show the distribution of the number of unique top coauthors for members of the latter group. In Fig. 1 (right panel) we show the probability for authors belonging to such groups of being a top-cited scientist themselves in their 20th career year.

The left panel reveals, as one would intuitively expect, a positive correlation between institutional prestige and coauthorship with top scientists. The right panel, in turn, shows a positive correlation between institutional prestige and the probability of becoming a top-cited scientist in the long run. Yet, regardless of the relative position in terms of institutional prestige, such probability is significantly higher for researchers who coauthored papers with one top scientist, and markedly higher for those who did so with more than one top scientist.

Furthermore, the right panel shows that, on average, the probability of becoming a top-cited scientist is below the aforementioned unconditional one (grey shaded area) for almost the entire pool of junior researchers lacking a top coauthor in their early career, with only those in the top quintile of institutional prestige managing to do better. Conversely, junior researchers who publish with top-cited scientists are in the opposite situation, and achieve better-than-average impact regardless of their position in terms of institutional prestige. In Supplementary Fig. 2 we show that patterns very similar to those in Fig. 1b are obtained when considering the citations accrued by the three groups of junior researchers throughout their career.

**Different dimensions of early career excellence.** We now proceed to expand this analysis by assessing how excellence in different aspects of academia relates with long-term impact by splitting all junior researchers in our pool into eight mutually exclusive groups based on early career performance according to different indicators. Namely, we consider institutional prestige (I), productivity (P), measured by the number of papers published within the first 3 career years, and the citations received within the first 3 career years (C). We group junior researchers depending on whether they belong to the top 10% of authors across such dimensions. (Authors are compared against their peers in the same discipline who started their career in the same year. In all cases where the top decile falls within a group of scientists with the same number of papers of citations, we only select those scientists whose number of papers or citations is strictly larger than the top decile. In Supplementary Table 5 we report the values of such thresholds for all disciplines and years.) For example, we label as I the group of researchers belonging to the top 10% in terms of institutional prestige, as IP (IC) the group of researchers belonging to the top 10% in both institutional prestige and productivity (citations), and as IPC the group of authors belonging to the top 10% of all three dimensions.

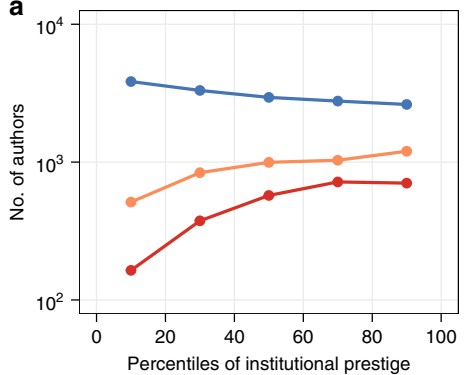
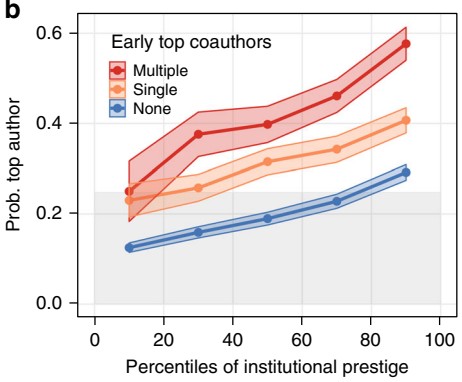

**Fig. 1** Relationship between early career institutional prestige and probability of becoming a top scientist. **a** Number of junior researchers in each quintile of the distribution of institutional prestige. **b** Probability of being a top scientist in the 20th career year as a function of institutional prestige (ribbon bands denote 95% confidence intervals). In both panels authors are grouped based on whether in their first 3 career years they coauthored papers with one (orange), multiple (red), or no (blue) top scientists. The grey shaded area in **b** represents the unconditional probability of becoming a top scientist for the entire pool of junior researchers

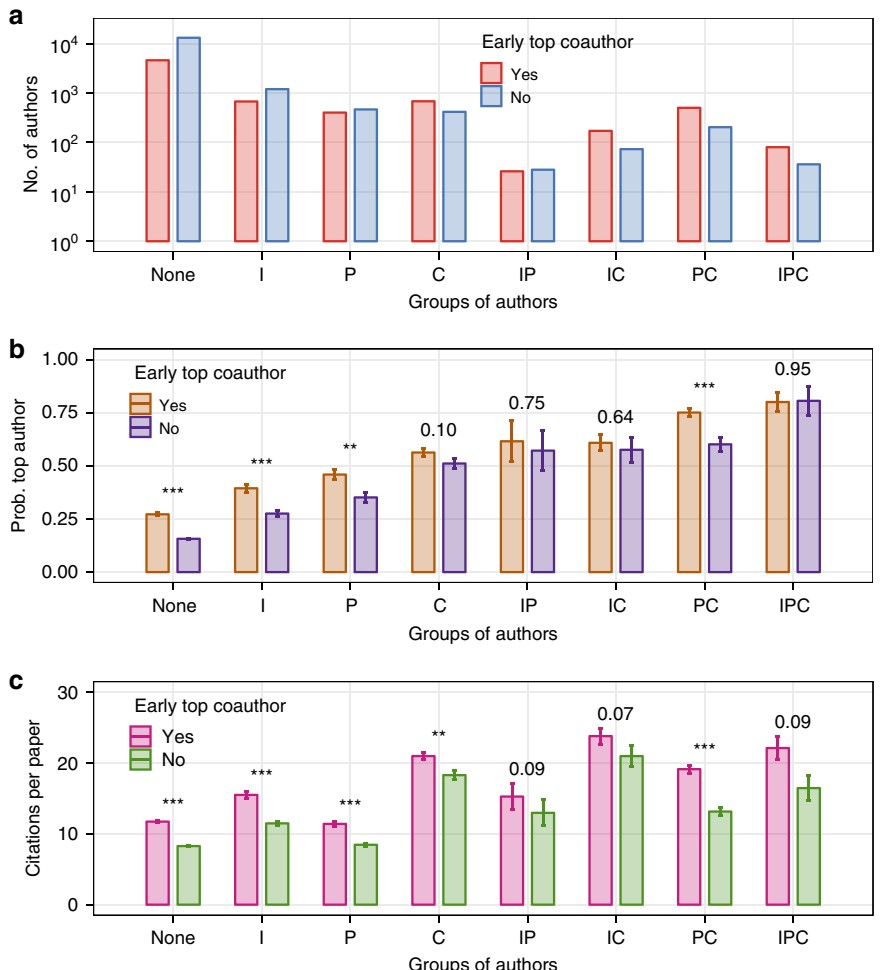

**Fig. 2** Relationship between long-term impact and early career performance. **a** Number of junior researchers belonging to the top 10% in various categories of early career performance (I denotes institutional prestige, P denotes productivity, and C denotes citations received. All three such quantities are computed based on the first 3 career years). **b** Probability for authors belonging to each group of being a top scientist in their 20th career year. **c** Number of citations received per paper published by authors belonging to each group between their 4th and 20th career year. In **b** and **c** we report 95% confidence intervals, and we report the p-values obtained via t-tests to assess the statistical significance of differences between the sub-group of junior researchers who coauthor work with a top scientist in the first 3 career years and the sub-group of those who do not. *$p < 0.05$; **$p < 0.01$; ***$p < 0.001$

The top panel in Fig. 2 shows, for each category, the number of junior researchers in our dataset who coauthored at least one paper with a top-cited scientist vs. those who did not. As it can be seen, the only group where the latter are the clear majority is the one of researchers who do not belong to the top 10% of any category. In all other cases, there is either a balance or a majority of junior researchers who coauthored with a top scientist, highlighting the presence of an overall positive correlation between coauthorship with top scientists and early career performance across all the dimensions we consider.

The middle panel shows the probability of becoming a top-cited scientist for authors belonging to each of the above categories. Overall, independently of coauthorship with top scientists, we find this probability to be progressively higher as we consider authors belonging to the top 10% of more categories, signalling a positive correlation between early and long-run career impact. Notably, such probability is above 50% for junior researchers belonging to the top decile of two dimensions (IP, IC, and PC), and hovers above 75% for junior researchers in the top decile of all three categories (IPC).

However, in all categories except the latter we find the above probability to be systematically higher for the sub-groups of junior researchers with an early career paper coauthored with a

top-cited scientist, and such differences are found to be statistically significant by a t-test in the cases labelled as "None", I, P, and PC. The results clearly show that the relative increase in the probability of becoming a top scientist tends to be larger in less exclusive groups, particularly in the group of junior researchers who do not belong to the top 10% of any of the categories considered. Indeed, for this group the coauthorship with a top scientist almost doubles such probability, which jumps from 15.7 to 27.2%. Large increases in the probability of becoming a top scientist are also apparent for the I and P groups. At the opposite end, coauthorship with a top scientist does not make a difference for junior researchers in the IPC group. One could interpret this as evidence that members of the latter group are with high probability already on the pathway to long-term career impact, regardless of their coauthors. In contrast, coauthorship with a top scientist truly has potential career-altering consequences for junior researchers who are not in the top 10% of any of the categories we considered. In the following, we elaborate more on the mechanics leading to such consequences.

The bottom panel in Fig. 2 shows analogous results in terms of citations received per paper published between the 4th and 20th career year. We observe similar patterns to those shown in the

middle panel, i.e. we find the sub-groups of junior researchers who coauthor with top scientists to systematically receive more citation than their peers in all categories. For the sake of readability, here we only show aggregate results. In Supplementary Figs. 3–6, we show equivalent figures for each of the four disciplines we consider.

In Supplementary Fig. 7 we show instead a breakdown of Fig. 2c, showing the citations received by the junior researchers between their 4th and 20th career year from papers published with and without top scientists as coauthors, in order to assess the contribution of the latter to the junior researchers' impact. In Supplementary Fig. 8, we specialise the latter case to each discipline. When aggregating all disciplines, we see again that those who coauthored work with top scientists in the first 3 years still achieve greater impact than those who did not, with statistically significant differences in the same group as in Fig. 2c, except for the one labelled as C. When considering individual disciplines, we still observe relevant differences between the junior researchers who coauthor work with top scientists and those who do not, with the former still typically attracting more citations per paper published than the former. However, in most disciplines such differences are statistically significant only for those sub-groups of junior researchers who belong to the top 10% of their field in just one or none of the dimensions considered (i.e. the sub-groups labelled as 'None', I, P, and C). This result suggests that the impact of early career coauthorship with a top scientist is somewhat inversely proportional to the impact already achieved by a junior researcher, and in the following we will demonstrate that this is indeed the case.

**Matched paired analysis**. The above results begin to reveal a systematic competitive advantage for junior researchers who coauthor with a top scientist when considered as a group, but do not yet quantify such advantage at the level of individual careers. Figuratively speaking, this could only be measured by tracking a young researcher in two parallel careers where all factors remain identical, except that in one she gets to write a paper with a top scientist and in the other she does not. This is akin to a medical trial situation, where the effectiveness of a new drug has to be assessed by forming a treatment and a control group.

We follow this line of reasoning and form two such groups in order to carry out a matched pair experimental design. Namely, in each of the disciplines considered we identify pairs of junior researchers with similar early career profiles in terms of institutional prestige, productivity, and impact (i.e. number of citations accrued), with the only difference being that only one of the two has coauthored a paper with a top scientist during her first 3 career years (we shall refer to this as treatment). We form such pairs via propensity score matching, following ref. [41] (see Supplementary Fig. 9). We then proceed to assess whether this has a detectable long-term effect by computing the average number of citations accrued between career years 4 and 20 by authors belonging to each group, both including and excluding those received by the papers published during the first 3 career years. In order to discount productivity as a possible confounding factor, we also compute the average number of citations received per paper published between career years 4 and 20. In particular, we focus on authors with low early career impact (i.e. with no more than ten citations received in the first 3 career years) in order to focus on the group of junior researchers who can benefit the most from the interaction with a top scientist. Overall, there are 2324 such authors in Cell Biology, 5635 in Chemistry, 5605 in Neuroscience, and 5414 in Physics.

The results of the analysis are reported in Table 1. In all four disciplines, we identify several hundreds of matched pairs of

**Table 1 Matched pair analysis results**

| | Cell Biology | | | Chemistry | | | Neuroscience | | | Physics | | |
|---|---|---|---|---|---|---|---|---|---|---|---|---|
| | Treat | Control | p | Treat | Control | p | Treat | Control | p | Treat | Control | p |
| No. of authors | 3176 | | | 6304 | | | 6439 | | | 6682 | | |
| No. of pairs | 468 | | | 1443 | | | 1602 | | | 1362 | | |
| Inst. prestige | 22.40 (0.57) | 22.24 (0.66) | NS (NS) | 26.85 (0.32) | 26.99 (0.39) | NS (NS) | 10.52 (0.17) | 10.66 (0.19) | NS (NS) | 19.58 (0.24) | 19.69 (0.27) | NS (NS) |
| Productivity | 2.36 (0.06) | 2.37 (0.06) | NS (NS) | 3.09 (0.04) | 3.13 (0.05) | NS (NS) | 3.13 (0.04) | 3.09 (0.04) | NS (NS) | 3.35 (0.05) | 3.25 (0.05) | NS (NS) |
| Cit. (years 1–3) | 4.80 (0.15) | 4.71 (0.14) | NS (NS) | 4.06 (0.08) | 3.95 (0.08) | NS (NS) | 3.95 (0.07) | 3.92 (0.07) | NS (NS) | 4.22 (0.08) | 4.19 (0.08) | NS (NS) |
| (A) | 361.91 (18.70) | 281.74 (15.13) | *** (***) | 301.36 (8.49) | 246.38 (7.17) | *** (***) | 383.13 (10.68) | 325.60 (10.73) | *** (***) | 394.28 (13.88) | 288.31 (9.47) | *** (***) |
| (B) | 315.78 (18.05) | 247.25 (14.62) | *** (**) | 259.68 (8.18) | 209.86 (6.80) | *** (**) | 314.56 (9.76) | 273.18 (9.95) | *** (**) | 343.09 (13.05) | 249.04 (8.92) | *** (***) |
| (C) | 14.39 (0.48) | 11.56 (0.44) | *** (***) | 7.61 (0.12) | 6.67 (0.12) | *** (***) | 12.23 (0.24) | 10.12 (0.20) | *** (***) | 10.14 (0.21) | 8.59 (0.22) | *** (***) |
| (D) | 3.61 (0.17) | 2.72 (0.16) | *** (***) | 4.85 (0.13) | 2.93 (0.09) | *** (***) | 3.49 (0.09) | 2.53 (0.08) | *** (***) | 4.29 (0.14) | 2.68 (0.10) | *** (***) |
| (E) | 5.37 (0.30) | 4.05 (0.30) | ** (***) | 10.71 (0.38) | 5.97 (0.25) | ** (***) | 6.91 (0.24) | 4.98 (0.22) | *** (***) | 9.73 (0.39) | 6.15 (0.32) | *** (***) |

Junior researchers are matched based on the institutional prestige they are embedded in, their productivity (measured by the number of papers published), and the number of citations received during their first 3 career years. One researcher per pair is either assigned to the treatment group (those who coauthored at least one paper with a top scientist) or the control group, and we compute the average of the following quantities across the two groups (numbers in brackets denote standard errors): (A) Citations received in career years 1–20. (B) Citations received in career years 4–20 excluding those received by the papers published in the first 3 career years. (C) Citations received per paper published during career years 4–20. (D) Number of different top scientists (per paper published) with whom the researcher has coauthored papers in career years 4–20 (excluding those already accounted in the first 3 career years for the treatment group). (E) Number of times (per paper published) the researcher has coauthored papers with a top scientist in career years from 4 to 20 (excluding those already accounted in the first 3 career years for the treatment group). The Significance levels shown refer to t-tests and Kruskal-Wallis tests (in brackets)
NS not significant
*p < 0.05; **p < 0.01; ***p < 0.001

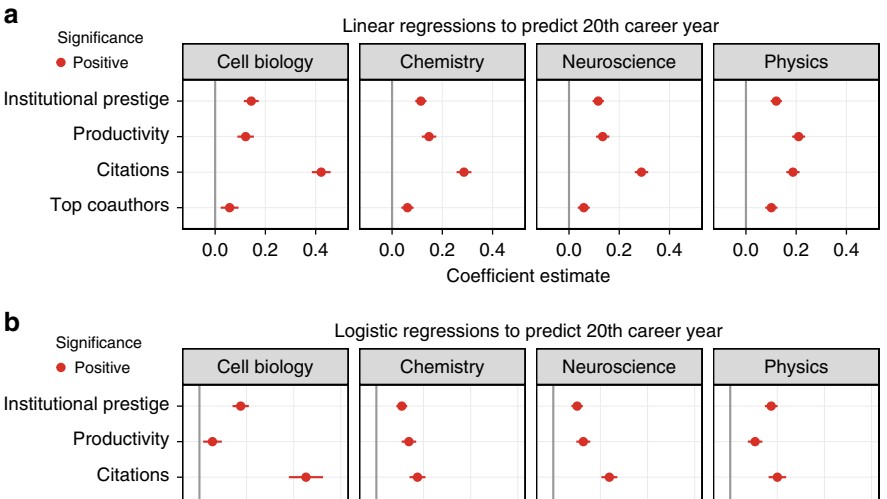

**Fig. 3** Regression analysis of long-term career impact. **a** Results from discipline-specific linear regressions whose dependent variable is the number of citations accrued in the first 20 career years ($R^2 = 0.33, 0.21, 0.18,$ and $0.19$ for Cell Biology, Chemistry, Neuroscience, and Physics, respectively). **b** Results from discipline-specific logistic regressions whose dependent variable is a binary indicator denoting whether a junior researcher is a top scientist in her 20th career year (AUC $= 0.785, 0.711, 0.726,$ and $0.749$ for Cell Biology, Chemistry, Neuroscience, and Physics, respectively). In all cases there is a statistically significant positive relationship between coauthorship with a top scientist in the first 3 career years and long-term impact. Bars refer to to 95% confidence intervals

junior researchers with similar early career profiles, except for the presence/lack of a top coauthor. In all disciplines we find the treatment group of junior researchers who coauthored with a top scientist to achieve a higher impact, regardless of the specific citation metric, and we find the differences with respect to the control group to be statistically significant in all cases, both when testing sample averages via $t$-tests and when testing the entire distributions via Kruskal–Wallis tests (in order to account for the skewness in the data, especially in the case of citations).

This result demonstrates the long-lasting competitive advantage associated with early career coauthorship with top scientists. In order to understand the mechanism through which such a competitive advantage materialises, we measure how often on average junior researchers belonging to the two above groups get to coauthor papers with top scientists between years 4 and 20 of their careers. The results from this analysis show that the treatment group consolidates its early competitive advantage by getting more opportunities to further collaborate with top scientists than the control group. This happens both in terms of the number of different top coauthors (excluding those already accounted for in the first 3 career years for the treatment group) and the number of individual coauthorship events with top scientists. Indeed, we find statistically significant differences between the treatment and control groups in all disciplines, with the former outperforming the latter in terms of repeated access to top scientists.

In Supplementary Table 6 we show that within pairs the junior researcher in the treatment group is the most cited in absolute terms ($p < 0.001$ in Chemistry, Physics, and Neuroscience, $p < 0.01$ in Cell Biology, one-tailed binomial test), and also the one who subsequently gets to coauthor more times with top scientists ($p < 0.001$ in Chemistry, Physics, and Neuroscience, $p = 0.38$ in Cell Biology, one-tailed binomial test). As an additional robustness control, in Supplementary Table 7 we show that the matched pair analysis results do not change when matching junior researchers based on their first 5 career years. Furthermore, in Supplementary Table 8 we report additional

results obtained when including the number of unique coauthors for publications during the first 3 career years as an additional covariate. We find our results to be qualitatively unchanged for the most part, although with reduced statistical significance in Chemistry and Neuroscience.

Put together, the above results suggest that coauthorship with a top scientist potentially represents a good predictor of impact in a long-lived academic career. This is confirmed by the outcomes of discipline-specific linear and logistic regressions, where we use early career coauthorship with at least one top scientist as a binary regressor against future impact, while controlling for institutional prestige, productivity, and impact in the first 3 career years (see the regression plot in Fig. 3). As dependent variables, we use the number of citations accrued in the first 20 career years in the case of linear regressions, and a binary variable to indicate whether a junior researcher had become a top scientist herself (i.e. among the top 5% cited scientists in her discipline) in her 20th career year in the case of logistic regressions, respectively. We systematically find coauthorship with at least one top scientist to be a statistically significant predictor of long-term future impact. Odds ratios for early collaboration with top coauthors in logistic regressions are: 1.19 for Cell Biology, 1.15 for Chemistry, 1.14 for Neuroscience, and 1.14 for Physics.

## Discussion

In this paper we presented a number of analyses to assess the effect of early career coauthorship with established top-cited scientists on the long-term prospects of junior researchers' academic impact. Invariably, our results highlighted that junior researchers who get the opportunity to coauthor at least one paper with a top scientist in the first few years of their career achieve a persistent competitive advantage with respect to their peers who do not get such an opportunity. Therefore, the following question becomes the crux of the matter: is such a competitive advantage a reflection of a young researcher's exceptional

skills, which in turn lead her to collaborate with a top scientist, or is it instead a direct consequence of the interaction with a top scientist?

Our results cannot provide a definitive answer to the above question for one fundamental reason, i.e. that we cannot control for the fact that established top scientists might attract the very best students. This acts as an ineradicable confounding factor in our analysis. Nevertheless, our results are systematic enough to suggest that the latter is the most plausible explanation, i.e. that the collaboration with a top scientist creates the aforementioned competitive advantage, whose echo can still be detected 20 years later.

Let us clarify that our results do not imply that all successful careers are launched thanks to the interaction with a top scientist early on. The probability to become a top scientist in the long run is the highest for those junior researchers who start their careers as the best among their peers (see Fig. 2). The fact that they excel in early career citations, productivity, and institutional prestige is enough to guarantee their long-lasting academic impact (see Fig. 3) independently of their coauthors. However, the coauthorship with top scientists truly makes the difference at lower strata of early career excellence: for those junior researchers who are not at the top among their peers in at least one category among institutional prestige, productivity, and impact, the opportunity to coauthor papers with top scientists systematically provides a competitive advantage that can unlock their potential and shift their career trajectory. This is further corroborated by the fact that junior researchers who do not belong to the top 10% of their field in any dimension but have the opportunity to coauthor work with top scientists keep attracting citations at a higher rate than their peers even for papers that are not coauthored with other top scientists throughout their entire career (see Supplementary Fig. 7). In this respect, our work sheds light on previously published studies on the interactions between junior and well-established scientists[42,43].

The aforementioned competitive advantage materialises by means of a "rich-get-richer" mechanism, where the early career opportunity to coauthor papers with a top scientist translates into a higher probability of doing it again at later career stages, and, eventually, to become one. In this respect, our results are in line with the long-standing observation that academic success breeds further academic success[44], which is empirically supported by a number of studies that have shown how academic achievements facilitate further impact and recognition[32,45].

The present work sheds new light on the determinants of academic impact. Indeed, our results show that early career opportunities can play an important role in shaping the prospects of a long academic career. Loosely speaking, we may say that being "in the right place at the right time"—and being able to seize on the opportunity—provides a junior researcher with an early edge, which may separate her from her peers for years to come. This is highlighted very clearly by our matched pair analysis (see Table 1), which shows that the interaction with a top scientist in the first 3 career years is already enough to permanently split career trajectories that were otherwise on the same path.

It is tempting to relate the above results to the "Newton hypothesis", i.e. the idea that Science mostly progresses thanks to the work of a few elite contributors who typically "stand on the shoulders of giants", i.e. who rely on the previous work of other elite scientists (as opposed to the "Ortega hypothesis", which instead purports that scientific progress mostly comes from the incremental contributions of many average scientists[46]). A number of studies on citation networks provide empirical support to the Newton hypothesis, showing that newly published papers tend to lean on a handful of important past contributions

in their field[47], and that disciplines are organised around "rich clubs" of top scientists that tend to preferentially cite their peers' work[48].

Our results also go in that direction, showing that scientific elites play an exceedingly important role in shaping the academic landscape at all its levels. As shown in Fig. 2, the early career coauthorship with a top scientist has the strongest effect—in relative terms—in the case of junior researchers who are not in the top among their peers in terms of productivity, impact, or academic prestige (i.e. the group labelled as "None" in the figure). The interaction with a top scientist still does not put this group on par with their peers who excel in some or all of these categories already at an early career stage (see, e.g. the group labelled as "IPC"), but nevertheless provides them with an opportunity to achieve visibility (and impact) that they seemingly could not achieve otherwise.

Our results complement those of[34], whose authors demonstrated that academic hiring networks are best explained by institutional prestige rather than meritocratic factors. Our findings support the idea that the most effective way for junior researchers in less prestigious institutions to escape such a "prestige trap" would be to connect with a top scientist in their field. Seen from a different angle, we interpret this as evidence that less prestigious academic institutions are filled with untapped potential. This, once combined with the evidence in[34] about current hiring academic practices, centred around a minority of top institutions and scientists, suggests that a significant portion of such potential may actually remain unrealised.

In line with other contributions[26,36,37], our findings suggest that citation counts are driven by multiple factors, including "social" ones, which makes it difficult to assess the intrinsic merit of an individual researcher just from the number of her citations. We hope that the present work will contribute to spur the development of nuanced bibliometric indicators, such as the equivalent of the "wins above replacement" metrics that are used to assess the contribution of individual players in team sports, or "visibility-adjusted" citation counts aimed at comparing more fairly the impact achieved by a scientist with respect to her peers with similar career trajectories.

As a final remark, we ought to acknowledge possible limitations in our study. First, our data do not allow us to identify cases of junior researchers coauthoring papers with top scientists from different disciplines. We reasonably expect such cases to be a very small minority, but their undetected presence may still have a small impact on some of our results. Second, due to the limited information available from the data used in this study, our analyses had to be performed at the aggregate level of entire disciplines, and could not be pushed to the level of individual fields of research. In this respect, applying automated topic identification techniques to the full text of papers would allow to investigate the relationship between junior researchers and top scientists more deeply by quantifying how much of the long-term impact of the former is achieved in the same sub-field of research of the latter. However, the consistency of our findings across the four disciplines we considered is encouraging in this respect, and we speculate that our results would still hold if tested at more granular scales.

## Methods

**Data.** We collected publication and citation data for four disciplines (Cell Biology, Chemistry, Neuroscience, and Physics) indexed on the Web of Science database. For each discipline, we collected data about all papers published since 1970 in a selection of journals, their authors and their affiliations. The data include outputs such as letters and editorials, but we limited our dataset to standard articles and review articles, as these are the usual outputs of research efforts.

We selected journals based on two criteria: in the case of Chemistry and Physics, we selected all publications issued by the American Physical Society (APS,

nine journals) and the American Chemical Society (42 journals), which represent major publishers in their respective fields. In the case of Cell Biology and Neuroscience, instead, we selected all journals whose publications collectively accrued at least 10,000 total citations according to the Journal Citation Reports. These amount to 53 journals in Cell Biology and 59 journals in Neuroscience, respectively. We provide the full lists of journals in Supplementary Tables 1–4.

We then proceeded to disambiguate the names of the authors of papers in the above venues with the methodology published in ref. [14], and we only retained the papers and citations belonging to authors with at least ten citations in their final career year. With these positions, we retained 226,362 papers and 71,794 authors in Cell Biology, 524,639 papers and 123,513 authors in Chemistry, 395,246 papers and 102,074 authors in Neuroscience, and 412,063 papers and 80,218 authors in Physics. We ought to acknowledge that cases of authors whose names change over time cannot be easily detected with the above methodology. However, we reasonably expect such cases to be a tiny fraction of the total number of authors in the dataset.

**Institutional prestige score**. We measure the institutional prestige a junior researcher is embedded in by means of the Nature Index (https://www.natureindex.com/), which has been introduced by the Nature group in order to rank the academic prestige of universities and research institutions. This is computed by counting the number of papers published in a set of expert-selected journals in the above four disciplines (Cell Biology, Chemistry, Neuroscience, and Physics). We adopt the same methodology and compute a given institution $i$'s prestige score as $\sqrt{P_i^{\text{nat}}}$, where $P_i^{\text{nat}}$ is the number of publications authored by researchers affiliated with institution $i$ in the Nature Index's list of journals since 1970. We then compute a paper's prestige score as the average prestige score of its authors' institutions, and a researcher's prestige score as the average prestige score of her papers.

**Reporting summary**. Further information on research design is available in the Nature Research Reporting Summary linked to this article.

## Data availability
The APS data used in the paper are publicly accessible and can be downloaded via https://journals.aps.org/datasets. The other publication and citation data are available via Web of Science (https://wok.mimas.ac.uk/).

## Code availability
The code for used to perform pair matching is available at https://cran.r-project.org/web/packages/MatchIt/index.html. All other codes used in this study are available from the corresponding author upon reasonable request.

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

## Acknowledgements

W.L. and G.L. were supported by an EPSRC Early Career Fellowship in Digital Economy (Grant No. EP/N006062/1).

## Author contributions

W.L., T.A., F.C. and G.L. designed and performed research, contributed new analytic tools, and wrote the paper. W.L. analyzed data.

## Competing interests

The authors declare that they have no competing interests.
