## [Transparent Peer Review File · Nature Communications]

Reviewers' comments:

Reviewer #1 (Remarks to the Author):

The topic of this paper is interesting, but how to prove that working with top scientists at the early career stage of young scholars can benefit their careers. It is a hard problem, oftentimes, there are too many confounding factors which are difficult to exclude or explain. Also citations are the biased measure as these citations might be brought by the top scientists, not the young scholars. The difference of citations between the papers with and without top scientists after the first 3 years should be analyzed for these two groups (one collaborate with top scientists during the first three years, one not).

More comments:

- please provide more details on matching process:

. in your supplementary file, table 1: do you mean that you match junior scholars based on the first 3 years: # of publications, # citations, and institute prestige, or # of citations of junior scholars are measured in 2018?

. Do you treat your three criterias equally which means that institution prestige counts the same as productivity, or citation, if that is the case, can you provide evidences to support your assumption

. how do you match, say for productivity: A has 10 papers, B has 15 papers, do you treat A and B the same or different? Or how do you deal with A has inst prestige 0.8, publications 10 and citations 20, while B has inst prestige 0.6, publications 15, and citations 25. Do you treat them same or different? Do you consider use propensity score matching to set up your pairs?

. what are other confounding factors, how could you exclude them, such as: the number of unique co-authors of junior scholars, the different research topics of junior scholars (given you use fields to mitigate the difference, but these fields are too large which can accommodate many different topics, say physics: quantum physics is different from complex network), can you please illustrate more details on confounding factors and how you treat them?

. In table 1 of your supplementary file, you want to provide evidences to show that first 3 years is a legitimate choice. Do you try first 5 years, as in US, most doctoral students take 5 years to graduate, and many did not publish articles in their first three years. It will be worthwhile to compare the difference whether first 3 or first 5 years are a better choice.

- please explain your choice of the list of journals you selected for each field? Any backup support on the list? Also why you choose these four fields, not other fields?

- Each field might have different number of junior scholars and top scientists, what are the ratio of them? Does some field have low ratio which means it is a lot harder for a junior scholar to be able to coauthor with a top scientist. How do you treat that difference in these four fields. Do you also consider a junior scholar co-authored with a top scientist in a different field?

- did you disambiguate author names? how do you trace those people in their 20th career year if their names might change or same for their affiliations (see manuscript Table 1 B). Actually in Table 1 B, the institutional prestige events play a bigger role than the coauthor with top-cited scientist to increase the probability of success of young scholars as the difference among these three categories increase significantly with the increase of institutional prestige. What will be an example for a junior scholar to co-author with more than one top-cited scientist during the first 3 years. Normally it is their doctoral study periods, the most likely is to work with their supervisor who is the top cited scholar. Can you please check more details with the group who co-authored with more than one top scientists, how many they co-author, how many pairs, are these common? What are the number of pairs in your multiple group, if the number of pairs are not in a comparable size of these three groups, the new way of comparison should be conducted to minimize the bias of sample size.

- set up the 3 year window for junior scholars to either collaborate with the top scientists or not might not be reasonable. As 3-year window is too arbitrary. Research shows that those young scholars who did not collaborate with the top scientists during their first 5 years, later on manage to collaborate with top scientists after the first 5 years are more successful than those who collaborated with the top scientists during their first 5 years. More in depth analysis and design are important to justify the statements you claimed in the first paragraph on page 5.

- In Figure 2, if I understand correctly, yes if the junior scholar collaborates with top scientists, no means not. Then A does not show the clear advantage of the difference between these two groups, with I, P, and none have more young scholars belong to top 10% without collaboration with top scientists. C part for citation per paper, which can be citations brought by the top scientists not the young scholar. The comparison here is biased. If you really want to compare the citation difference, you might want to see papers with top scientists and papers without top scientists, for these two groups.

Reviewer #2 (Remarks to the Author):

In my opinion, the paper under discussion reports interesting research results. Technically, the paper is in good shape. Theoretically, however, the paper is poorly linked to previous literature. The author repeatedly emphasize that this field of studies has "recently" emerged and that this data is only recently made available. The authors fail to relate to an existing literature since Robert K. Merton's development of a sociology of science in the late-1960s including a relation to existing citation data (Elkana, Lederberg, Merton, Thackray, & Zuckerman, 1978). The other pioneer to be mentioned is the historian Derek de Solla Price (e.g., Price, 1965, 1976). In the field of bibliometrics/scientometrics many of the choices discussed here have been debated.

For example, the mechanism identified here can be positioned in relation to the debate of the so-called Ortega hypothesis picturing the sciences as "icebergs" vs. the so-called Newton hypothesis that the "giants" stand on the shoulders of "giants" forming an elite layer in the sciences with also its own reproduction mechanism. The authors provide support for the latter hypothesis, but interpret their results perhaps erroneously in the light of the former when they state on p. 10 that "[w]e interpret his result as evidence that academic institutions are filled with untapped potential, which largely remain unrealized simply due to a lack of opportunity." The problem is not "simply due to a lack of opportunity" because this "lack of opportunity" is due to the same root cause; namely, the mechanism of recruitment by the scientific elite.

The lack of reading of the relevant literature weakens, in my opinion, also the operationalization by disregarding or accepting limitations. For example, the impact factor or other, but similar considerations can be expected to have played a role in the journal selections. Similarly, the author choose for Nature index when measuring institutional status; I would have chosen for the Leiden Rankings of research universities given the objectives of the study. But the authors may have good reasons for their choices. In the present text, choices seem to be a bit ad-hoc and pragmatic; perhaps, also in relation to the absence of reflection of the theoretical sources that could have been consulted. I am aware that in footnote 24, the authors refer to Garfield's (2006) discussion of the impact factor, but that is only the top of another iceberg which is, in my opinion, insufficiently unpacked in this contribution.

For example, the skewness of scientometric distributions. The authors mention this and use quintiles for the analysis of the prestige distributions. However, the authors do not proceed to non-parametric statistics for the testing of differences (Kruskal-Wallis), but use t-tests of differences between means. I am not a statistician, but I would like to hear arguments why to make these seemingly obvious choices despite the arguments in the literature questioning these choices. Perhaps, eventually this may make no difference and the conclusions still hold. Perhaps, the authors define their "new" science of science as not burdened with these discussions in the past

decades, while seemingly able to reach out to (and convince?) a relatively “lay” audience which is unfamiliar with the technical and theoretical details and assumptions. From my perspective, however, these issues have to be raised.

Where does this bring me in terms of an advice to the Editor? In my opinion, some major revisions are due. Some of these revisions may lead to other parameter choices. However, the computation is not the problem. Using other choices, the results may be different and perhaps more nuanced.

References

Elkana, Y., Lederberg, J., Merton, R. K., Thackray, A., & Zuckerman, H. (1978). *Toward a Metric of Science: The advent of science indicators*. New York, etc.: Wiley.

Price, D. J. d. S. (1965). Networks of scientific papers. *Science*, 149(no. 3683), 510- 515.

Price, D. J. d. S. (1976). A general theory of bibliometric and other cumulative advantage processes. *Journal of the American Society for Information Science*, 27(5), 292-306.

Reviewer #3 (Remarks to the Author):

This is a well-written and well-researched study that proves how junior researchers co-authoring with top scientists in their early careers get a competitive advantage compared to junior researchers not co-authoring with top scientists in their early careers. However, it is of course difficult to prove a causal relationship between the mere co-authoring with top scientists and later career development. The authors are well aware of this, and relate to this (page 9) in the discussion. This is a basic condition for all bibliometric studies: it is often possible to prove correlations, but difficult/impossible to prove causal relations.

The first paragraph of the paper lacks some more history. Starting with the sentence “The availability of data about published research has led academia to increasingly study itself over the last few years” somewhat bothers an old (and proud) bibliometrician like myself. Bibliometrics has a long and important history of quantitative science studies that unfortunately is often neglected. I think the authors should consider broadening the first paragraph and include a few historical examples of bibliometric studies or at least refer to a few literature reviews to show a better understanding of the history of the field (and that it goes a long way back!). A bit in line with this is the mentioning of the impact factor (page 2) as an easy impact measure. Yes, it is an easy measure, but it is also an extremely criticized measure – not only by academics, but also repeatedly by bibliometricians who have invented many alternative impact measures. Again, as I am a bibliometrician, and as I have worked in the field for more than 20 years, such a sentence “The first two factors are somewhat easier to measure, thanks to the availability of multiple indices aimed at ranking journals (e.g., the impact factor) and institutions” simply hurts my ear. However, it is quite common to read those kind of statements in non-bibliometric journals. I don’t think you would get away with such statements in specialized journals. Bibliometricians know about all the problems and issues concerning using various proxies for measuring quality, impact, activity, scholarliness, etc., etc. That said, I am fully aware that this is a paper for a non-specialized journal, and that all such “technicalities” cannot be dealt with here. Yet, I think the authors should consider a bit more restricted language, and be a bit more “humble”, acknowledging the important research done by bibliometricians over many years exactly showing the difficulty in accurately measuring quality, impact, activity, scholarliness, etc., etc.

All in all I think it is a very interesting study, and I have no problems with the methods being used.

Response to the reviews of manuscript NCOMMS-19-18447: “Achieving competitive advantage in academia through early career coauthorship with top scientists”

We thank all three Reviewers for their comments and feedback, which gave us the opportunity to substantially improve our paper. We have extensively rewritten several sections, particularly the Introduction and Discussion in order to make better contact between our work and the Bibliometrics literature. In the following, we reply to all comments from the Reviewers in the same order as they appear in their reports.

Response to Reviewer 1

Please provide more details on matching process:

- *In your supplementary file, table 1: do you mean that you match junior scholars based on the first 3 years: # of publications, # citations, and institute prestige, or # of citations of junior scholars are measured in 2018?*
- *Do you treat your three criterias equally which means that institution prestige counts the same as productivity, or citation, if that is the case, can you provide evidences to support your assumption*
- *How do you match, say for productivity: A has 10 papers, B has 15 papers, do you treat A and B the same or different? Or how do you deal with A has inst prestige 0.8, publications 10 and citations 20, while B has inst prestige 0.6, publications 15, and citations 25. Do you treat them same or different? Do you consider use propensity score matching to set up your pairs?*

Addressed: Let us address these questions by starting from the last point. Indeed, we carried out our matching analysis by forming pairs via propensity scores. In particular, we numerically implemented the methodology outlined in Ho, et al., “Matching as non-parametric preprocessing for reducing model dependence in parametric causal inference”, *Political Analysis* (2007), which is an established benchmark in the literature on pair matching. In this respect, we have therefore treated the three covariates in our analysis (number of publications, number of citations, institutional prestige) with equal importance. We have added a reference to the above paper in the Results section, and we have added propensity score plots as Supplementary Figure 7 in order to show that the control and treatment groups in our analysis (i.e., junior researchers who have / do not have

written a paper with at least one top scientist in their first three career years) have very close means in all three covariates across the whole spectrum of propensity score values.

Concerning the first point in the above list, we match junior researchers based on the number of citations received within the first three career years in order to discount possible time misalignments (which would instead be a factor if we considered the citations received towards the end of the available data).

What are other confounding factors, how could you exclude them, such as: the number of unique co-authors of junior scholars, the different research topics of junior scholars (given you use fields to mitigate the difference, but these fields are too large which can accommodate many different topics, say physics: quantum physics is different from complex network), can you please illustrate more details on confounding factors and how you treat them?

Addressed: We selected productivity, impact and institutional prestige as the covariates for our pair matching analysis based on *i*) their widely recognized importance as determinants of career success in academia (as mentioned in the Introduction of the paper), and *ii*) their straightforward quantifiability in terms of papers published, citations accrued, and institutional rankings (indeed, the ranking of institutions based on the Nature Index, which we use as proxy for prestige, strongly correlates with many of the available published rankings. We added a mention to this in the Results section, see also a reply to Reviewer 2 below).

We chose to carry out our analyses in terms of broadly defined disciplines (such as Physics, Chemistry, etc.) for two main reasons. First, this choice is in line with most of the recently published literature on scientific impact and success (this is indeed the case for most of the references cited in our bibliography). Second, there is no straightforward and commonly accepted way to quantify the proximity between two authors in terms of research field. To the best of our knowledge, this has been done either by quantifying the overlap between the keywords that are normally associated with published papers or in terms of the similarity of cited research (see, e.g., Ciotti, et al., *EPJ Data Science* (2016)). Yet, both methods have limitations that would prevent from conducting a large-scale and multidisciplinary analysis, such as the one we performed in our paper. Namely, the former method would require working with a very limited set of publication venues in order to have some consistency in terms of labelling conventions for different sub-fields of research. Similarly, the latter method would drastically reduce the number of journals and papers that could be used for the analysis, as it requires the full bibliographies of all papers employed. Such data are not made systematically available, and retrieving them (e.g., via scraping) would go well beyond the scope of this paper. We have added a paragraph

at the end of the Discussion section to acknowledge this as a possible limitation to our study.

Following the Reviewer's suggestion, we have expanded our matched pair analysis to include the number of unique coauthors (during the first 3 career years) as an additional control covariate. The results are reported in Supplementary Table 6. As it can be seen, adding this further control does not produce significant qualitative changes in the results for Cell Biology, Physics, and Chemistry (albeit in the latter case, the pair matching procedure we use leads to statistically significant differences between the treatment and control groups in two control covariates, i.e., the numbers of citations and unique coauthors). In the case of Neuroscience, instead, 4 out of 5 of the variables we test still display statistically significant differences when measured in the treatment and control groups, but this is no longer the case for the citations received in career years 4-20 when removing those obtained from papers published in the first 3 years. We have added a comment to these results in the Results section of the main text.

In table 1 of your supplementary file, you want to provide evidences to show that first 3 years is a legitimate choice. Do you try first 5 years, as in US, most doctoral students take 5 years to graduate, and many did not publish articles in their first three years. It will worthwhile to compare the difference whether first 3 or first 5 years are a better choice.

Addressed: This is a very good point. Indeed, we had neglected to fully justify our reasoning behind the choice of running our analyses on the first 3 career years. In fact, what we mean by that is that we consider the first 3 years after the publication of the first paper. Reasonably, we expect the end of this period to be around the 4th or 5th year since the start of the PhD. We have added a comment in the first paragraph of the Results section to clarify this.

In addition, following the Reviewer's suggestion, we have repeated our pair matched analysis on the first 5 career years (where, again, by that we mean 5 years after the publication of the first paper). We have reported the results of this new analysis in Supplementary Table 6 and mentioned them in the Results section of the paper. All the results (and their significance) reported in the original analysis are confirmed with no exceptions.

Please explain your choice of the list of journals you selected for each field? Any backup support on the list? Also why you choose these four fields, not other fields?

Addressed: We used two different selection criteria in order to test whether our results would be robust with respect to possible changes in the data. As detailed in the Data

section of the paper, we chose papers in Chemistry and Physics based on their publisher. Namely, we selected all journals issued by the American Chemical Society and the American Physical Society, which represent the largest publishers in their respective fields. On the other hand, in the other two disciplines that we considered (Cell Biology and Neuroscience) we chose journals based on an impact-related criterion, i.e., we retained all journals whose papers collectively accrued at least 10,000 citations over their lifetime according to the Journal Citation Reports (JCR).

Each field might have different number of junior scholars and top scientists, what are the ratio of them? Does some field have low ratio which means it is a lot harder for a junior scholar to be able to coauthor with a top scientist. How do you treat that difference in these four fields. Do you also consider a junior scholar co-authored with a top scientist in a different field?

Addressed: In Fig. 1 below we report the ratios between top scientists and junior researchers in each of the four disciplines we consider, computed annually from 1980 to 1998 (i.e., the period in which we select the junior researchers for the analyses in our paper). As it can be seen, such ratios are rather stable around $\sim 10\%$ and are very similar across the four disciplines. For the time being, we have not included the plot in Fig. 1, but we would be happy to do so if the Reviewer felt it would add valuable information to the paper.

Figure 1: Annual ratios between the numbers of top scientists and junior researchers in each discipline.

Coming to the second part of the above question, the answer is unfortunately no. The data are organised in terms of journals and disciplines, so determining whether some junior researcher's coauthor is a top scientist in a different field would require scanning the entirety of available published papers in search for that particular coauthor, which would be unfeasible. We added a mention to this limitation of our study at the end of the Discussion section.

Did you disambiguate author names? How do you trace those people in their 20th career year if their names might change or same for their affiliations (see manuscript Table 1 B).

Addressed: As mentioned in the Data section of the paper, we disambiguated names using the method published in Sinatra et al., *Science* (2016), which provides one of the most reliable options in the literature. To the best of our knowledge, no available disambiguation method can reliably track authors whose names change over time (e.g., due to changes in marital status). This is an intrinsic limitation that cannot be avoided without additional (and specific) information on the authors of interest. Yet, we are reasonably confident that only a very small fraction of the authors in our dataset might be “lost” in our analysis due to name changes. We have added a mention of this point in the Data section of the paper. Conversely, changes in institution do not represent a problem, and authors can still be reliably tracked based on their name even after changing affiliations.

Actually in Table 1 B, the institutional prestige events play a bigger role than the coauthor with top-cited scientist to increase the probability of success of young scholars as the difference among these three categories increase significantly with the increase of institutional prestige. What will be an example for a junior scholar to co-author with more than one top-cited scientist during the first 3 years. Normally it is their doctoral study periods, the most likely is to work with their supervisor who is the top cited scholar. Can you please check more details with the group who co-authored with more than one top scientists, how many they co-author, how many pairs, are these common? What are the number of pairs in your multiple group, if the number of pairs are not in a comparable size of these three groups, the new way of comparison should be conduct to minimize the bias of sample size.

Addressed: As the Reviewer points out, the top scientists associated with junior researchers in the first 3 or so career years can be reasonably expected to be their PhD

supervisors. To put their relevance to our analyses in perspective, in our dataset we have a total of 22,601 junior researchers, 7,106 of which have coauthored work with at least one top scientist. Within this group, 2,533 (11.2% of the sample, 35.6% of the junior researchers with at least one top coauthor) have coauthored work with multiple top scientists (2 or 3 in the vast majority of cases, see Fig. 2 below). We have added these numbers in the paper and we have included the plot below as Supplementary Figure 1.

Figure 2: Distribution of the number of unique top coauthors for junior researchers in the four disciplines.

We are not entirely sure what the Reviewer meant in the second part of the above question, i.e., the point concerning pairs. We did not use the number of top coauthors as a covariate in our matched pair analysis precisely because the vast majority of junior researchers with at least one top coauthor have exactly one top coauthor. We treated the presence of top coauthors as a binary event, i.e., we pooled all junior researchers with *at least* one top coauthor to form the treatment group. In this respect, we are not sure what the Reviewer means when referring to three groups in relation to the matched pair analysis. We show three different groups (no top coauthor, 1 top coauthor, multiple top coauthors) in Fig.1 of the paper to show that the presence of multiple top coauthors seems to provide additional competitive advantage in the long run, but then we merge the two latter groups when performing the matched pair analysis. We hope this clarifies any issues the Reviewer had in mind. If not, we would of course be happy to do so.

Set up the 3 year window for junior scholars to either collaborate with the top scientists or not might not be reasonable. As 3-year window is too arbitrary. Research shows that

those young scholars who did not collaborate with the top scientists during their first 5 years, later on manage to collaborate with top scientists after the first 5 years are more successful than those who collaborated with the top scientists during their first 5 years. More in depth analysis and design are important to justify the statements you claimed in the first paragraph on page 5.

Addressed: We thank the Reviewer for raising the issue that our choice of a 3 years window for coauthorship with top scientists was to be better discussed and tested for robustness. As explained in our response to an earlier comment, we now clarify in the paper our reasoning behind this choice and we tested the robustness of our results with respect to changes in the length of the time window (see Supplementary Table 6). Concerning the fact that authors can coauthor papers with top scientists also beyond this time window and benefit from it: our question was if and to what extent junior scientists can benefit from coauthorship with top scientists. Our answer to that question is that yes, it is beneficial to co-author papers with a top cited scientist early in one's career, especially for those junior scientists who come from less prestigious universities. This does not rule out the fact that non-junior scientists can also benefit from interaction with top scientists. In fact our analysis suggests that those authors who coauthor a paper with a top scientist as junior scientists will also coauthor more papers with top scientists as non-junior scientists with respect to their peers who never coauthored a paper with a top scientist while being junior scientists.

In Figure 2, if I understand correctly, yes is the junior scholar collaborates with top scientists, no means not. Then A does not show the clear advantage of the difference between these two groups, with I, P, and none have more young scholars belong to top 10% without collaboration with top scientists. C part for citation per paper, which can be citations brought by the top scientists not the young scholar. The comparison here is biased. If you really want to compare the citation difference, you might want to see papers with top scientists and papers without top scientists, for these two groups.

Addressed: We thank the Reviewer for this comment, which prompted us to further unpack the relationship between early and long-term career impact. Following the Reviewer's suggestion, we have added Supplementary Figure 6, which breaks down panel C of Fig. 2 to the papers published with and without top coauthors. Namely, panel A (B) of the new Supplementary Figure shows the number of citations received per paper published by junior researchers between years 4 and 20 of their careers with (without) top scientists as coauthors. As it can be seen, the patterns in both panels are very similar to those obtained in the aggregate case of Fig. 2, panel C. Namely, in all categories the authors who had a top scientist as coauthor during their first three career years enjoy a systematic

competitive advantage, with no exceptions. Such advantage is statistically significant in the groups labeled as “None”, I, P, and PC, regardless of whether the papers between career years 4 ad 20 are published with or without a top scientist. The only difference with respect to the aggregate case presented in the main paper is the group labeled as C (i.e., junior researchers who are in the top 10% in their field in terms of citations accrued in the first 3 career years), which does not show a statistically significant difference between junior researchers who published work with / without a top scientist, as is instead the case after aggregation.

Response to Reviewer 2

In my opinion, the paper under discussion reports interesting research results. Technically, the paper is in good shape. Theoretically, however, the paper is poorly linked to previous literature. The author repeatedly emphasize that this field of studies has ‘recently’ emerged and that this data is only recently made available. The authors fail to relate to an existing literature since Robert K. Merton’s development of a sociology of science in the late-1960s including a relation to existing citation data (Elkana, Lederberg, Merton, Thackray, & Zuckerman, 1978). The other pioneer to be mentioned is the historian Derek de Solla Price (e.g., Price, 1965, 1976). In the field of bibliometrics/scientometrics many of the choices discussed here have been debated. For example, the mechanism identified here can be positioned in relation to the debate of the so-called Ortega hypothesis picturing the sciences as “icebergs” vs. the so-called Newton hypothesis that the “giants” stand on the shoulders of “giants” forming an elite layer in the sciences with also its own reproduction mechanism. The authors provide support for the latter hypothesis, but interpret their results perhaps erroneously in the light of the former when they state on p. 10 that “we interpret his result as evidence that academic institutions are filled with untapped potential, which largely remain unrealized simply due to a lack of opportunity.” The problem is not “simply due to a lack of opportunity” because this “lack of opportunity” is due to the same root cause; namely, the mechanism of recruitment by the scientific elite.

Addressed: We concur with the Reviewer that our original submission was lacking in terms of its acknowledgment of previous literature and in terms of the interpretation of our findings in relation to ongoing debates. We realise that we ended up highlighting more recent papers at the expense of pioneering contributions. Following the Reviewer’s suggestions, we have considerably amended both the Introduction and Discussion sections of our paper.

We have rewritten the beginning of the former in order to acknowledge more of the Bibliometrics literature and some of its earliest contributors, and to clarify that recent data-driven contributions (those we referred to as the “Science of Science”) do not represent a novel field but rather a complement to a preexisting one.

Similarly, we have extensively changed the final part of the Discussion section in order to address the issue raised by the Reviewer concerning the debate around the Newton and Ortega hypotheses. We agree that the language in the previous version of the paper was potentially confusing, and we have now clarified that our results mostly support the Newton hypothesis. We also have better clarified our previous remarks on less prestigious institutions representing reservoirs of untapped potential. We concur with the Reviewer that the lack of realisation of such potential is another manifestation of the very same phenomenon (elite scientists recruiting preferentially from elite institutions), and we have written so explicitly, highlighting the role played by the coauthorship with top scientist as a pathway to escape the “prestige trap” in which the majority of junior researchers in less prestigious institutions find themselves in.

The lack of reading of the relevant literature weakens, in my opinion, also the operationalization by disregarding or accepting limitations. For example, the impact factor or other, but similar considerations can be expected to have played a role in the journal selections. Similarly, the author choose for Nature index when measuring institutional status; I would have chosen for the Leiden Rankings of research universities given the objectives of the study. But the authors may have good reasons for their choices. In the present text, choices seem to be a bit ad-hoc and pragmatic; perhaps, also in relation to the absence of reflection of the theoretical sources that could have been consulted. I am aware that in footnote 24, the authors refer to Garfield's (2006) discussion of the impact factor, but that is only the top of another iceberg which is, in my opinion, insufficiently unpacked in this contribution.

Addressed: We thank the Reviewer for giving us the opportunity to clarify our operational choices. To begin with, we have added clearer justifications for our definitions of “top scientists” and “junior researchers” (see the first paragraph of the Results section).

Concerning the point raised by the Reviewer about rankings, we completely agree that published university rankings, such as the Leiden ones, represent the ideal quantifiers of academic prestige. Yet, from a technical standpoint, our approach based on pair matching requires to assign some form of institutional prestige “score” to each junior researcher in the dataset, i.e., a real number to quantify (both in absolute and in relative terms) the overall prestige the researcher is embedded in. This is the motivation that initially led us towards the Nature Index, which allowed us to do that as explained in the Data section of

the paper. However, our choice was further corroborated by the fact that when ranking institutions based on their Nature Index, one obtains results that are exceedingly close to most of the regularly published rankings. For example, the Kendall correlation coefficients between the Leiden rankings and the rankings obtained from the Nature Index are 0.98 for Cell Biology, 0.94 for Chemistry, 0.94 for Neuroscience, and 0.97 for Physics. We have added comments on these points in the Results section of the paper.

We also concur with the Reviewer that our previous brief mention of Garfield's discussion on the impact factor was insufficient to convey the complexity inherent to quantifying the impact of different publication venues, and the controversy associated with it. We have added a new paragraph to the Introduction (see page 2) in order to mention those important points.

Let us also mention that in preliminary versions of the analyses reported in the paper, we had considered using the average impact factors of a junior researcher's early career publication as an additional covariate in our matched pair analysis, but ultimately found that it was very strongly correlated with our proxy for institutional prestige.

For example, the skewness of scientometric distributions. The authors mention this and use quintiles for the analysis of the prestige distributions. However, the authors do not proceed to non-parametric statistics for the testing of differences (Kruskall-Wallis), but use t-tests of differences between means. I am not a statistician, but I would like to hear arguments why to make these seemingly obvious choices despite the arguments in the literature questioning these choices. Perhaps, eventually this may make no difference and the conclusions still hold. Perhaps, the authors define their "new" science of science as not burdened with these discussions in the past decades, while seemingly able to reach out to (and convince?) a relatively "lay" audience which is unfamiliar with the technical and theoretical details and assumptions. From my perspective, however, these issues have to be raised.

Addressed: We thank the Reviewer for raising this point, which we had not considered. We had sought to complement the *t*-test results presented in Table 1 of the paper with those in Supplementary Table 5, where we had reported results of one-tailed binomial tests to measure how often members of the treatment group outperformed the peers they are matched with in our analysis. However, we do realize this might not still be sufficient evidence to support our claims, and therefore proceeded to complement our initial results with Kruskal-Wallis tests, as suggested by the Reviewer. The results are now shown in Table 1 of the paper: as it can be seen, they are fully in agreement with our initial findings, showing that the treatment group enjoys a systematic competitive advantage in the long run with respect to the control group in all 5 dimensions we consider. In addition to the

changes to the Table, we have also added a comment in Section XX to support the use of the Kruskal-Wallis test.

Response to Reviewer 3

The first paragraph of the paper lacks some more history. Starting with the sentence “The availability of data about published research has led academia to increasingly study itself over the last few years” somewhat bothers an old (and proud) bibliometrician like myself. Bibliometrics has a long and important history of quantitative science studies that unfortunately is often neglected. I think the authors should consider broadening the first paragraph and include a few historical examples of bibliometric studies or at least refer to a few literature reviews to show a better understanding of the history of the field (and that it goes a long way back!). A bit in line with this is the mentioning of the impact factor (page 2) as an easy impact measure. Yes, it is an easy measure, but it is also an extremely criticized measure not only by academics, but also repeatedly by bibliometricians who have invented many alternative impact measures. Again, as I am a bibliometrician, and as I have worked in the field for more than 20 years, such a sentence “The first two factors are somewhat easier to measure, thanks to the availability of multiple indices aimed at ranking journals (e.g., the impact factor) and institutions” simply hurts my ear. However, it is quite common to read those kind of statements in non-bibliometric journals. I don’t think you would get away with such statements in specialized journals. Bibliometricians know about all the problems and issues concerning using various proxies for measuring quality, impact, activity, scholarliness, etc., etc. That said, I am fully aware that this is a paper for a non-specialized journal, and that all such “technicalities” cannot be dealt with here. Yet, I think the authors should consider a bit more restricted language, and be a bit more “humble”, acknowledging the important research done by bibliometricians over many years exactly showing the difficulty in accurately measuring quality, impact, activity, scholarliness, etc., etc.

Addressed: We thank the Reviewer for his/her comments. As commented in the response in Reviewer #2, it was certainly not our intention to neglect decades of work by Bibliometrician. We realize that our initial list of references gave too much importance to recent data-driven developments. We have extensively amended both our Introduction and our bibliography in order to provide a more accurate historical account of the origins of Bibliometrics and its earliest fundamental contributions.

Similarly, we expanded the paragraphs devoted to mentioning, e.g., bibliometric indicators and the impact factor. Again, it was not our intention to “hide” the complexity

associated with measuring the quality and impact of academic outputs (or the issues associated with the use of different proxies).

In addition, also following feedback from Reviewer 2, we expanded our final discussion to frame our findings within long-standing debates in Bibliometrics and the history of Science. Namely, we have added some comments on how our results relate to the long-standing debate on the Newton vs Ortega hypotheses, arguing that our findings mostly support the former.

Reviewers' comments:

Reviewer #1 (Remarks to the Author):

The results you have proven are devastating, as the mere co-authorship with top scientists during the early stage of the career can already decide a final fate of a junior scholar's long career, leaving no space and hope for them to change. It is hard to face, but it is reality according to your data and research. I appreciate your efforts of doing such and also question myself whether there are some ways to uncover positive output that hardworking can somehow change the fate. This also triggers another thought that whether practicing science is a social activity that social bias exists (such like this, which is a kind of bias that elites hire elites, and elites have more resources and chances to reach the top) since citation (which is the measure here) is a social behavior which can not equal to science correctness or scientific truth. We oftentimes hear stories that a discovery from an unknown remote institution took much longer to get the acceptance by the elites in the scientific society even it has strong proof to be true or effective (for example, Barry Marshall and Robin Warren of Perth from West Australia who found that ulcers are caused by *Helicobacter Pylori* bacterium.) I was wondering whether citation is the right measure in general. Your research provides another evidence that citation might not be the right measure for scientific discovery and so called scientific success. In business, big corporations can be disrupted by small start-ups, competition seems fair albeit some social or political biases. But if we use citation as the measure for science, this case is clearly a biased case that no matter how many great things the young scholars have been done in their late careers, sorry, your early 3 year career will decide your future. I feel something is unjustified, but could not pinpoint the exact things on how to do. Maybe authors can have some discussions in their conclusion to call for better scientific measurement.

In page 5, Figure 1 A shows that institutions with high prestige also have lots of authors never co-authored with top scientists during their early career years. What are the fates of these guys. Do you think it is possible to fix one variable (say, they are all from the same level of institutions), one has co-authored with top scientists during the first 3 years of their career, the others not, then to see whether their citation measure changes during 4-20 years? Just curious that whether the co-authorship with top scientists in the early career has so much defining power? Or whether the paper co-authored with the top scientists brought more citations in the junior scholars' late career? How about the paper authored by the same junior scholar who co-authored with top scientists during their first 3 years, but this paper was not co-authored with the top scientists and topic of this paper is different compared with the topic of paper which was co-authored with top science (for topic similarity of two papers, you can use LDA to calculate the topic distribution for each paper, and use cosine similarity to see whether two papers are topically similar or not), what are the fates of these two different papers authored by the same young scholar? Or the late citations are more or less coming from the papers with co-authored top scientists, which show the accumulative advantage of collaborating with the top scientists. In this case, I do not think it is the success of the young scholar, but the continuous success of his/her coauthors (these top scientists), either being the real top scientists or being the fake top scientists because of their supervisors (means that no new ideas being generated during the young scholars' late career).

P is productivity for young scholars during their first 3 years. I wonder that number should be very low, say 2-3 papers in total? can you show us the real numbers of publication and also what is the threshold of being top 10% of P (means how many articles they need to publish in order to be able to get to the top 10%). Same for C, I guess that c is also very small. How do you choose or define top 10%. Say you have 3 published 10 papers, 8 published 5 papers, and 30 published 4 papers. Suppose, your top 10% is $3+8+2=13$, how do you select 2 from 30 authors who published 4 papers?

in your Figure 2 B from supplementary, IP and IPC the probability to become top scientists are much higher than those who co-authored with top scientists during their first 3 years. It is very interesting, can you please elaborate more on that? Especially IP that authors without co-

authoring with top scientists have much high (almost double the other group) to be top scientists later on. Very exciting, worth more exploring, can you please provide some cases from your data? For C, two groups seem to have similar citations per paper on C, but quite big difference if you add I and P. It is interesting, can you please explain? It seems that it does not matter whether you coauthor with top scientist early on or not, you still can reach almost the same number of citations as the one who co-authored with top scientists. I think it is an exciting result as the citation accumulated from group (not) is much harder than group (yes) as group Yes has the accumulative advantage from their co-authored top scientists. Same results for Figure 3 B in chemistry, in IPC, NO group has much higher probability of being top scientists than Yes group. I was wondering whether prestige of institute and productivity have more defining power than co-authoring with top scientists? I think this makes better sense to me and worth further explanation. Same for Figure 5 C, the No group is almost the same as the Yes group given IPC for citations per paper.

Figure 6 A and B, it is clear that if without the papers co-authored with top scientists, these two groups (yes and no groups) are similar in citation per paper (in B), while in A, they are quite different. It shows the accumulative advantage brought by the top scientists, not the young scholars themselves. This I will say fake fame. Again, it is an exciting result and worth to further explore.

All these above evidences show that whether co-authored with top scientists during your early career years cannot decide your future success (as measured getting more citations).

This is already much-improved version. But I encourage authors to go further to identify hidden patterns, rather just sit on the conclusion that co-author with top scientists during your early career can automatically bring the success to you no matter whether you as the young scholar work hard or being innovative.

Reviewer #2 (Remarks to the Author):

I endorse publication of this revision. Thank you so much.

Loet Leydesdorff
University of Amsterdam

Reviewer #3 (Remarks to the Author):

I am glad to see that the authors now have explicitly recognized the long history of the field. I think this suits the paper, and I have no problems with accepting it as it is.

Response to the reviews of manuscript NCOMMS-19-18447: “Achieving competitive advantage in academia through early career coauthorship with top scientists”

We thank Reviewers #2 and #3 for recommending the publication of our revised manuscript, and we thank Reviewer #1 for his/her positive assessment and the additional feedback, which has given us the opportunity to further improve our paper. In the following, we reply to all comments by Reviewer #1 in the same order as they appear in his/her report, and we detail the corresponding changes we have made in the manuscript.

Response to Reviewer 1

The results you have proven are devastating, as the mere co-authorship with top scientists during the early stage of the career can already decide a final fate of a junior scholar's long career, leaving no space and hope for they to change. It is hard to face, but it is reality according to your data and research. I appreciate your efforts of doing such and also question myself whether there are some ways to uncover positive output that hardworking can somehow change the fate. This also triggers another thought that whether practicing science is a social activity that social bias exists (such like this, which is a kind of bias that elites hire elites, and elites have more resources and chances to reach the top) since citation (which is the measure here) is a social behavior which can not equal to science correctness or scientific truth. We oftentimes hear stories that a discovery from an unknown remote institution took much longer to get the acceptance by the elites in the scientific society even it has strong proof to be true or effective (for example, Barry Marshall and Robin Warrent of Perth from West Australia who found that ulcers are caused by Helicobacter Pylori bacterium.) I was wondering whether citation is the right measure in general. Your research provides another evidence that citation might not be the right measure for scientific discovery and so called scientific success. In business, big corporations can be disrupted by small start-ups, competition seems fair albeit some social or political biases. But if we use citation as the measure for science, this case is clearly a biased case that no matter how many great things the young scholars have been done in their late careers, sorry, your early 3 year career will decide your future. I feel something is unjustified, but could not pinpoint the exact things on how to do. Maybe authors can have some discussions in their conclusion to call for better scientific measurement.

Addressed: We thank the Reviewer for this comment. We concur with the Reviewer that citations (or citation-based bibliometric indicators, such as the *h*-index) may not be reliable indicators of the quality of a scientist's work, and we believe our work contributes

to show that citations may partially reflect social factors that have not necessarily to do with it, such as a scientist's visibility and that of his/her coauthors. Nevertheless, papers and citations are the only two *observable* outputs in published science, and any quantitative study of academic impact necessarily has to rely on them in one way or another. We believe, however, that our results provide novel insight to the debate around how to quantify academic impact in fair ways. Following the Reviewer's suggestion, we have added a paragraph to our Discussion where we mention this point and call for the development of 'visibility-adjusted' bibliometric indicators aimed at providing fair comparisons between the impact achieved by a scientist and those among his/her peers who had similar career trajectories.

Let us also stress that, as we will discuss in response to the other comments of the Reviewer, our results show that co-authorship with top scientists does not affect all junior researchers equally, as on average it tends to benefit more those researchers who do not enjoy much success in their early career in the other dimensions we consider (early career citations, productivity, and institutional prestige). Indeed, as we show in Fig. 3 of the main paper, such dimensions are better predictors of future impact than the early career coauthorship with top scientist, whose impact becomes instead much more apparent when explicitly controlling for all such factors (as done in our matched pair analysis, see Table 1). We have added a few remarks about this point in our Discussion.

In page 5, Figure 1 A shows that institutions with high prestige also have lots of authors never co-authored with top scientists during their early career years. What are the fates of these guys. Do you think it is possible to fix one variable (say, they are all from the same level of institutions), one has co-authored with top scientists during the first 3 years of their career, the others not, then to see whether their citation measure changes during 4-20 years? Just curious that whether the co-authorship with top scientists in the early career has so much defining power? Or whether the paper co-authored with the top scientists brought more citations in the junior scholars' late career?

Addressed: In order to further investigate the fates of scientists belonging to the groups in Fig. 1A, we have produced plots with the citations received in years 1-20 vs. those received in years 4-20 for the same groups of authors in Fig.1A (Supplementary Figure 2). This analysis follows the Reviewer's suggestion as each point in the plots corresponds to a given quantile of institutional prestige, which is then varied along the x axis. The new plots show that there is a clear separation between the three groups (junior researchers with no top coauthors, with one top coauthors, and with more than one top coauthor) in terms of citations received, which persists if we restrict the number of citations to those accrued in the years 4-20.

Further insight on the Reviewer's questions can also be extracted from the matched pair analysis presented in Table 1 of the main paper. Indeed, such analysis precisely relies on matching pairs of authors whose early career trajectories are very similar (in terms of institutional prestige, early career citations and productivity) except for the co-authorship with a top scientist. Row A of the Table reports the total number of citations received on average by members of the two groups (where 'Treatment' denotes junior researchers who coauthored at least one paper with a top scientist, and 'Control' denotes those who did not) during their first 20 career years, while row B reports the citations accrued only from papers published in years 4-20 (therefore excluding any paper coauthored with top scientists during the first 3 years for the Treatment group). In both cases the Treatment group maintains a clear and statistically significant competitive advantage in terms of total citations, which is also found when considering the average number of citations received per paper published (row C).

Concerning the final point, rows D and E of Table 1 shows that the competitive advantage of the Treatment group materialises as a greater probability to coauthor more often with other top scientists during the rest of their career (years 4-20). In this respect, as hinted by the Reviewer, it is arguably true that junior researchers who collaborate with top scientists early on in their career keep receiving citations from work coauthored with other top scientists at a higher rate than their peers. However, this is especially true for junior researchers who are not among the top early career scientists in their discipline.

How about the paper authored by the same junior scholar who co-authored with top scientists during their first 3 years, but this paper was not co-authored with the top scientists and topic of this paper is different compared with the topic of paper which was co-authored with top science (for topic similarity of two papers, you can use LDA to calculate the topic distribution for each paper, and use cosine similarity to see whether two papers are topically similar or not), what are the fates of these two different papers authored by the same young scholar? Or the late citations are more or less coming from the papers with co-authored top scientists, which show the accumulative advantage of collaborating with the top scientists. In this case, I do not think it is the success of the young scholar, but the continuous success of his/her coauthors (these top scientists), either being the real top scientists or being the fake top scientists because of their supervisors (means that no new ideas being generated during the young scholars' late career).

Addressed: Computing the distance between papers in terms of topics would be very interesting, but it is unfortunately not possible with the data available to us, which do not include the papers' full text. We added a mention to this as a possible future avenue of research at the end of our Discussion.

In order to address the Reviewer's comment, we have in any case added a new plot (Supplementary Figure 8), where we report the results of Fig. 2C of the paper after removing all citations received from work coauthored with top scientists. As it can be seen, in all disciplines there is still a very statistically significant difference between the Treatment and Control groups (labeled as 'Yes' and 'No', respectively) among the early career researchers who are not in the top 10% of any of the dimensions considered (the sub-group labeled as 'None'). In addition, depending on the discipline, there are other statistically significant differences in most sub-groups associated with early career excellence in one dimension only (i.e., the sub-groups labeled as I, P, and C). We have also added some comments to this point in the Results Section of the paper. We believe these results (in addition to those already presented in Table 1) are sufficient to clarify the role of top scientists. Indeed, Supplementary Figure 8 shows that even after the removal of all citations received from work coauthored with top scientists throughout their whole career, junior researchers in the lower strata of early career excellence (the sub-groups labeled as 'None', I, P, and C) still enjoy a clear competitive advantage with respect to their peers if they co-author a paper with a top scientist. As discussed in our conclusions, we believe this to be a reflection of the difference that top scientists can make in unlocking the potential of such junior researchers. Conversely, when removing the citations from work coauthored with top scientists, some of the differences in groups associated with higher excellence (IP, IC, PC, and IPC) are reduced. This is in line with our previous results, and with the regression results reported in Fig. 3 of the main paper.

P is productivity for young scholars during their first 3 years. I wonder that number should be very low, say 2-3 papers in total? can you show us the real numbers of publication and also what is the threshold of being top 10% of P (means how many articles they need to publish in order to be able to get to the top 10%). Same for C, i guess that c is also very small. How do you choose or define top 10%. Say you have 3 published 10 papers, 8 published 5 papers, and 30 published 4 papers. Suppose, you top 10% is 3+8+2=13, how do you select 2 from 30 authors who published 4 papers?

Addressed: We thank the Reviewer for allowing us to clarify the above point. In all cases where the top decile falls within a group of scientists with the same number of papers or citations, we only select those scientists whose number of papers or citations is strictly larger than that. We have added a footnote about this in the main paper, and a Table in the SI (Supplementary Table 5) to detail the values of all decile thresholds that we have used for each discipline and for each year of our analysis.

In your Figure 2 B from supplementary, IP and IPC the probability to become top scientists are much higher than those who co-authored with top scientists during their first 3 years. It is very interesting, can you please elaborate more on that? Especially IP that authors without co-authoring with top scientists have much high (almost double the other group) to be top scientists later on. Very exciting, worth more exploring, can you please provide some cases from your data? For C, two groups seem to have similar citations per paper on C, but quite big difference if you add I and P. It is interesting, can you please explain? It seems that it does not matter whether you coauthor with top scientist early on or not, you still can reach almost the same number of citations as the one who co-authored with top scientists. I think it is an exciting result as the citation accumulated from group (not) is much harder than group (yes) as group Yes has the accumulative advantage from their co-authored top scientists. Same results for Figure 3 B in chemistry, in IPC, NO group has much higher probability of being top scientists than Yes group. I was wondering whether prestige of institute and productivity have more defining power than co-authoring with top scientists? I think this makes better sense to me and worth further explanation. Same for Figure 5 C, the No group is almost the same as the Yes group given IPC for citations per paper.

Addressed: As the Reviewer mentions, indeed institutional prestige and productivity have greater predictive power, as shown by our regression results in Fig. 3 of the main paper. It is only when similar careers are aligned (as in our matched pair analysis or as done in Fig. 2 of the main paper and the equivalent ones in the SI with respect to early career excellence) that long-term impact of an early career coauthorship with a top scientist becomes apparent. In other words, we have to distinguish between two effects. At the *macroscopic* level, i.e. when considering all dimensions and all authors together (as in our regression analysis), then coauthorship with a top scientist is a statistically significant predictor of long-term academic impact, but a less powerful one than institutional prestige, early career productivity or early career impact (see Fig. 3 of the main paper). At the *microscopic* level of individual careers, instead, when controlling for all other dimensions and comparing *peers* with very close early career trajectories, then the impact of coauthorship with top scientists becomes a true difference-maker. We have commented on this in the Discussion.

The results mentioned by the Reviewers in the cases of Cell Biology and Chemistry (now supplementary Figures 3 and 4, respectively) are special cases of the above big picture at the macroscopic level. In both disciplines the early career coauthorship leads to statistically significant differences in impact for the 'None' group, i.e., authors who do not excel in any of the dimensions we consider. In contrast, when considering junior researchers who do excel in two or all three of such dimensions, then the predictive power

of the early career coauthorship with top scientists gets 'diluted' by the greater predictive power of those other covariates.

Figure 6 A and B, it is clear that if without the papers co-authored with top scientists, these two groups (yes and no groups) are similar in citation per paper (in B), while in A, they are quite different. It shows the accumulative advantage brought by the top scientists, not the young scholars themselves. This I will say fake fame. Again, it is an exciting result and worth to further explore.

Addressed: As already discussed in relation to another response above, we have added Supplementary Figure 8 to further unpack the relationship between the long-term impact of early career coauthorship with top scientists and the impact that other similar coauthorship events have throughout the rest of a researcher's careers. Some discussion in this respect has been added in the Results Section of the paper.

REVIEWERS' COMMENTS:

Reviewer #1 (Remarks to the Author):

Authors have provided the sufficient evidences and satisfied answers to my questions.